# The 2008–2010 Subsidence of Dallol Volcano on the Spreading Erta Ale Ridge: InSAR Observations and Source Models

**Maurizio Battaglia** [1,2,*] **, Carolina Pagli** [3] **and Stefano Meuti** [1]

1 Department of Earth Sciences, Sapienza University of Rome, 00185 Rome, Italy; stefs_@hotmail.it
2 U.S. Geological Survey, Volcano Disaster Assistance Program, Moffet Field, Mountain View, CA 94035, USA
3 Department of Earth Sciences, University of Pisa, 56126 Pisa, Italy; carolina.pagli@unipi.it
* Correspondence: mbattaglia@usgs.gov

**Abstract:** In this work, we study the subsidence of Dallol, an explosive crater and hydrothermal area along the spreading Erta Ale ridge of Afar (Ethiopia). No volcanic products exist at the surface. However, a diking episode in 2004, accompanied by dike-induced faulting, indicates that Dallol is an active volcanic area. The 2004 diking episode was followed by quiescence until subsidence started in 2008. We use InSAR to measure the deformation, and inverse, thermoelastic and poroelastic modelling to understand the possible causes of the subsidence. Analysis of InSAR data from 2004–2010 shows that subsidence, centered at Dallol, initiated in October 2008, and continued at least until February 2010 at an approximately regular rate of up to 10 cm/year. The inversion of InSAR average velocities finds that the source causing the subsidence is shallow (depth between 0.5 and 1.5 km), located under Dallol and with a volume decrease between $-0.63$ and $-0.26 \times 10^6$ km$^3$/year. The most likely explanation for the subsidence of Dallol volcano is a combination of outgassing (depressurization), cooling and contraction of the roof of a shallow crustal magma chamber or of the hydrothermal system.

**Keywords:** volcano deformation; interferometric synthetic aperture radar; ground deformation modelling; volcano geodesy; Dallol volcano

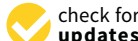



## 1. Introduction

Volcanoes commonly subside during, or after, eruptions as magma flows out of a chamber, or during post-eruptive periods when subsidence can occur because of cooling of magma, outgassing from the magma chamber, or viscoelastic relaxation of a spherical shell surrounding the chamber [1–4]). Cyclic uplift-subsidence periods have been explained as sudden inflows and successive cooling of magma, or they have been attributed to hydrothermal activity [5–8]. However, continuous subsidence which is not triggered by an eruption or an episode of deformation is more difficult to explain. The Askja volcano in Iceland has been subsiding for years without any eruption and the subsidence has been attributed to magma drainage, or cooling and crystallization of magma [9,10]. At Quaternary silicic calderas like Campi Flegrei (Italy) or Yellowstone (USA), the subsidence has been attributed to mixed magmatic–hydrothermal interaction [11,12].

Afar is a late-stage continental rift accommodating the divergence between the Arabian, Nubian, and Somalia plates along three rift arms: the Red Sea, the Gulf of Aden, and the Main Ethiopian rifts. Volcanism in Afar is focused on en echelon magmatic rift segments [13]. Dyke injections along the rift axis occur together with eruptions at central volcanoes. The Dallol magmatic rift segment is in the northern part of the Erta Ale ridge, where the Red Sea ridge moves inland creating the Afar continental rift (Figure 1). Extension rates in northern Erta Ale are relatively low, ~7 mm/year, increasing progressively to 20 mm/year further south in central Afar. The Dallol segment is a plain below sea level, filled by salt deposits. The Dallol explosive crater corresponds to a small topographic

high rising ~50 m above the salt plain. The associated hydrothermal system is the lowest-elevation subaerial volcano-hydrothermal system on Earth. Hydrothermal features include brightly colored hot springs that are dramatic. Seasonal brine springs are active in the plain around the Dallol crater, depositing salt crusts that have been mined for hundreds of years. No volcanic deposits outcrop at the surface; however, a ~20 m layer of basalt/dolerite was found in a borehole drilled in the salt plain around Dallol at 170 m beneath the surface [14], testifying to the presence of unerupted magma bodies at shallow depth. Indeed, the discovery of an active magma plumbing under Dallol, and its recognition as a nascent volcano with a rift segment was due to an InSAR study that captured a dyke intrusion in 2004 along the rift segment south of Dallol, fed by a magma chamber at 2.5 km depth [15].

In this paper, InSAR measurements were used to probe the deformation at Dallol. We show that since 2004 Dallol has been quiet, until continuous subsidence started in 2008 without any clear triggering. The satellite-based, remote sensing technique interferometric synthetic aperture radar (InSAR) was used to measure ground motions at Dallol. InSAR provides an all-weather, day/night capability to collect measurements, hence it has an edge over other remote sensing techniques such as multispectral imaging. Therefore, InSAR has been extensively used to detect ground motions at active volcanoes since the launch of the first European SAR satellites in the early 1990s by ESA (European Space Agency) (i.e., [4,7–11,15]). InSAR missions collect measurements with high spatial resolution (~20-by-20 m pixel) and temporal resolution, with revisit times ranging from ~one month for previous European satellites (ERS1/2 and Envisat) to a few days for the recent missions of Sentinel 1a/b. Time-series analysis also allows accuracy in estimating ground motion of only a few mm/year [16–19].

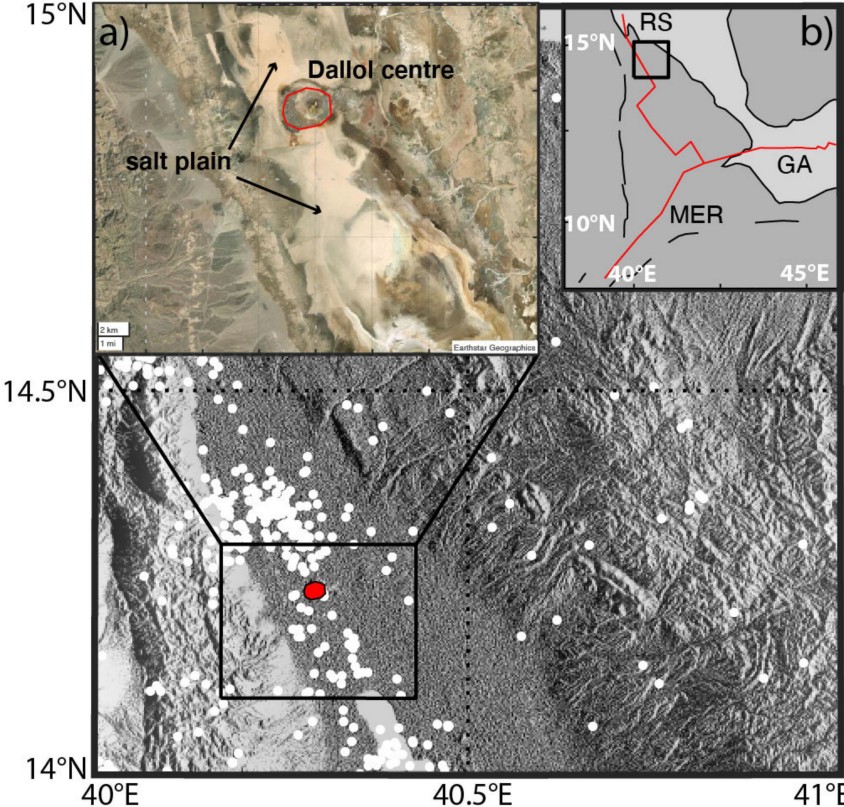

**Figure 1.** The main figure shows the area of Dallol. The red oval is the Dallol explosive crater and hydrothermal area (https://volcano.si.edu/volcano.cfm?vn=221041, accessed on 17 May 2021), and the white filled circles are earthquakes from [20]. (**a**) Panchromatic satellite image of the Dallol area. (**b**) Afar rift—the red line marks the rift axis and black lines are the rift-bounding faults. RS-Red Sea rift, GA-Gulf of Aden rift and MER-Main Ethiopian rift. The black box marks the location of the main figure.

## 2. InSAR Processing and Time-Series

We processed a series of interferograms from SAR images, acquired by the ENVISAT satellite between 2004 and 2010 in ascending and descending orbits, with the JPL/Caltech ROI_PAC software [21]. We used an external 3-arc sec (~90 m resolution) SRTM DEM for topographic correction. We filtered the interferograms using a power spectrum filter, before unwrapping them with a branch-cut algorithm. The final geocoding employed the same 3-arc sec SRTM DEM used for topographic correction [18].

The interferograms show no significant deformation in Dallol from 2004 until October 2008 when a concentric pattern, consistent with subsidence centered at Dallol, started, and continued at least until the end of ENVISAT acquisitions in early 2010. We selected the interferograms spanning the time of the subsidence at Dallol and inverted these to obtain maps of average surface velocities, and incremental time-series of displacements along the satellite Line-of-Sight (LOS), using the pi-rate time-series analysis program [18–20], (Figure 2). In the time-series analysis, any eventual residual unwrapping error was identified by using a phase closure method on Minimum Spanning Trees [18]. We applied orbital filtering to the geocoded interferograms by fitting them with a linear function, using a network approach [18]. We removed topographically correlated atmospheric noise [22] and applied the Atmospheric Phase Screen (APS) filter with a Gaussian temporal high-pass filter with 1σ of 0.5 years, to minimize atmospheric disturbances [23]. Finally, the interferograms were inverted for average velocity, incremental time-series of displacement and RMS error maps, using a linear least-square inversion with Laplacian smoothing. The temporal and spatial correlation between the interferograms were accounted for through the variance–covariance matrix [16,17,23]. We removed the unstable pixels that were not coherent in at least ten independent epochs and noisy pixels with an RMS misfit larger than 4 mm/year.

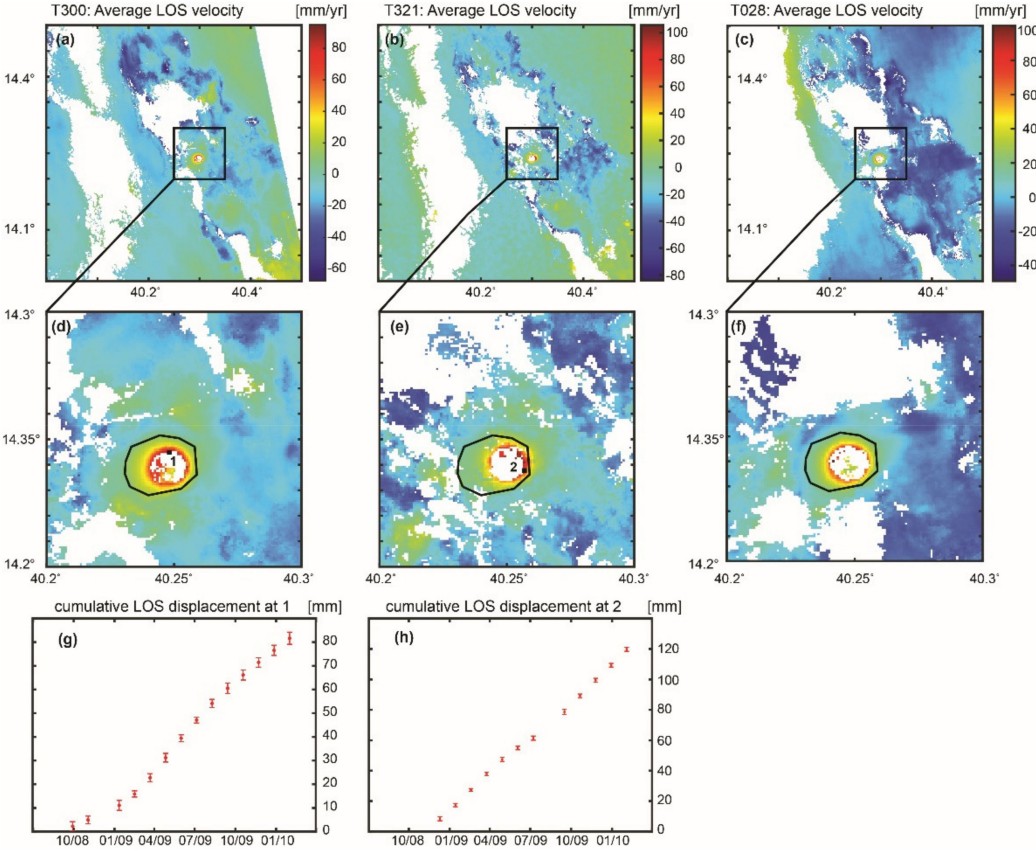

**Figure 2.** InSAR line-of-sight (LOS) average velocity maps of deformation at Dallol crater and surrounding areas from ENVISAT data. In this map, a LOS increase implies a subsidence. T300: ascending orbit; T321: descending orbit; T028: ascending orbit.

(**a–c**) InSAR average LOS velocity maps from 2008 to 2010, showing subsidence (increase in the LOS length) in the topographic depression of Dallol; the box marks the area shown in (**d–f**). (**d–f**) The black line marks the approximate boundary of the topographic depression marking the Dallol crater and hydrothermal area. The volcano formed by the intrusion of basaltic magma into the salt plains of Dallol (Figure 1), a vast area of uplifted thick salt deposits affected by intense fumarolic activity (https://volcano.si.edu/volcano.cfm?vn=221041 accessed on 09 April 2021). (**g,h**) LOS cumulative displacements at the numbered pixels in (**d,e**). No significant deformation is measured before October 2008 and after January 2010.

The average velocity maps from three different tracks exhibit good coherence at Dallol, though some noise occurs in the salt plain. All the maps show a consistent circular subsidence (LOS range increase, Figure 2a–f) centered at the Dallol crater of up to 100 mm/year during 2008–2010. The pattern is focused on a small area with a diameter of 4 km indicating that the source causing the Dallol subsidence is shallow. The pattern of cumulative displacement around Dallol (Figure 2g,h) shows that the rate of subsidence was approximately linear during 2008–2010 without any significant seasonal fluctuation.

The salt plain around the subsidence at Dallol crater is generally incoherent probably because of seasonal salt deposition. However, where coherence is maintained a signal of range decrease (line-of-sight uplift) is observed (Figure 2a–c). This signal is only seen over the salt plain. Therefore, we exclude any correlation to tectonic or volcanic processes. It is likely to be a radar propagation artefact caused by changes in backscattering coefficient in areas of high soil salinity and moisture [24].

## 3. InSAR Modelling

We tested different deformation mechanisms to explain the subsidence at Dallol, including reservoir contraction due to magma cooling as well as depressurization of the hydrothermal system.

### 3.1. Models of Reservoir Contraction

The InSAR deformation velocities were inverted using the dMODELS software package ([25]; https://pubs.usgs.gov/tm/13/b1/, accessed on 17 May 2021). dMODELS implements analytical solutions of different types of deformation sources embedded in a homogeneous, elastic half space. The software inverts for the best-fit model parameters from surface deformation data. For the InSAR inversion a broad range of source types commonly used to approximate magma chambers was assumed: a spherical source with correction for the topography [26,27], a spheroidal source [28], and two types of magma chambers and dikes: a horizontal penny-shaped crack [29] and a tensile dislocation [30]. The inversion scheme implements a weighted least-squares algorithm combined with a random search grid to infer the minimum of the penalty function [31]. Measurement errors are coded in the covariance matrix and the penalty function is the chi-square per degrees of freedom, $\chi_v^2$. The local minimum of the penalty function $\chi_v^2$ is determined using a constrained, nonlinear, multivariable interior-point algorithm (function *fmincon*, [32]). The data were decimated using a regular sub-sampling scheme. We employ a sub-sampling step equal to 2, corresponding to an inversion of 50% of the pixels of the InSAR image (Table 1; Figure 3).

The quantitative analysis to determine the best fit model is based on the comparison between the semivariograms of models ($\gamma_m$) and dataset ($\gamma_d$) (Table 1; Figure 4). The semivariogram is an essential tool in any geostatistical analysis [33]. It provides a graphical and numerical measure of the spatial distribution of the InSAR dataset and the results from the different models. By using the semivariogram, the comparison between the fit of different models to the experimental dataset is not based on the value of a single number, the $\chi_v^2$, but on the ability of the model to mimic the spatial distribution of the data (Figure 4).

**Table 1.** Summary of modeling results. Number of random searches 256. Selection radius for data set: 10 km from center of Dallol crater. Regular sub-sampling.

| Data Set | Description | Orbits | # of Pixels | Source | # of PARAM-ETERS | Orbits | $X_v^2$ | RMSE (1) Semi Variogram | ΔX0 (2)m | ΔY0 (2)m | Depthm b.s.l. | Radiusm | ΔV $10^6$ m³/Year |
|---|---|---|---|---|---|---|---|---|---|---|---|---|---|
| DALLOL 1 | Original data set | T321 &T300 | 78,467 | Sphere | 5 | T321 & T300 | 14.1 | 0.31 | 1647 | 841 | 24,437 | 172 | 5.29 |
| | | | | | | T321 & T028 | 18.2 | 0.28 | 1604 | 726 | 13,827 | 69 | 0.001 |
| | | | | Spheroid | 8 | T321 & T300 | 13.0 | 0.32 | −205 | 215 | 14,814 | 1228 | 45.3 |
| | | | | | | T321 & T028 | 15.4 | 0.30 | 1630 | 821 | 17,443 | 1049 | 46.2 |
| | | T321 &T028 | 77,008 | Penny-shaped crack | 5 | T321 & T300 | 14.0 | 0.87 | 1720 | 912 | 805 | 128 | 0.009 |
| | | | | | | T321 & T028 | 18.0 | 0.27 | 1720 | 912 | 5849 | 169 | 0.45 |
| | | | | Dike (3) | 8 | T321 & T300 | 10.4 | 0.22 | 1502 | 139 | 2684 | - | −0.93 |
| | | | | | | T321 & T028 | 11.6 | 0.20 | 461 | 912 | 5416 | - | −3.10 |
| DALLOL 2 | Reference point defined such that average deformation far away from crater is zero | T321 &T300 | 79,188 | Sphere | 5 | T321 & T300 | 10.4 | 0.12 | −96 | −92 | 1234 | 556 | −0.56 |
| | | | | | | T321 & T028 | 13.6 | 0.15 | −143 | −124 | 1277 | 60 | −0.59 |
| | | | | Spheroid | 8 | T321 & T300 | 10.3 | 0.10 | −112 | −127 | 1274 | 287 | −0.63 |
| | | | | | | *T321 & T028 (4)* | *13.4* | *0.63* | *−53* | *561* | *21,083* | *1103* | *−117* |
| | | T321 &T028 | 75,864 | Penny-shaped crack | 5 | T321 & T300 | 10.3 | 0.13 | −168 | −108 | 1470 | 1560 | −0.62 |
| | | | | | | T321 & T028 | 13.9 | 0.17 | −183 | −168 | 1241 | 1628 | −0.53 |
| | | | | Dike (3) | 8 | T321 & T300 | 11.8 | 0.20 | 63 | −278 | 551 | - | −0.48 |
| | | | | | | *T321 & T028 (4)* | *11.0* | *0.26* | *1526* | *815* | *7023* | *-* | *−2.68* |
| DALLOL 3 | Masked (only data that show subsidence) | T321 &T300 | 6127 | Sphere | 5 | T321 & T300 | 5.1 | 0.09 | −32 | −71 | 915 | 449 | −0.30 |
| | | | | | | T321 & T028 | 4.2 | 0.07 | −79 | −135 | 1046 | 463 | −0.32 |
| | | | | Spheroid | 8 | T321 & T300 | 5.0 | 0.10 | −83 | −77 | 798 | 582 | −0.31 |
| | | | | | | T321 & T028 | 4.2 | 0.08 | −82 | −142 | 1037 | 159 | −0.39 |
| | | T321 &T028 | 17,075 | Penny-shaped crack | 5 | T321 & T300 | 4.8 | 0.12 | −166 | −196 | 566 | 1430 | −0.26 |
| | | | | | | T321 & T028 | 3.8 | 0.12 | −216 | −223 | 516 | 1557 | −0.26 |
| | | | | Dike (3) | 8 | T321 & T300 | 3.9 | 0.06 | 605 | 305 | 707 | - | −0.29 |
| | | | | | | T321 & T028 | 3.6 | 0.09 | 764 | 196 | 724 | - | −0.28 |

step: 2 (inverted of 50% of data). T321: descending orbit. T300, T028: ascending orbits. (1) Normalized root mean square error of the fit between the variograms for models and data. (2) Source offset relative to center of crater: [640,251, 1,574,845] (UTM coordinates, zone 37P). (3) Location of midpoint of the dike. (4) We think this solution is an outlier: a valid mathematical solution with no geological meaning (italics).

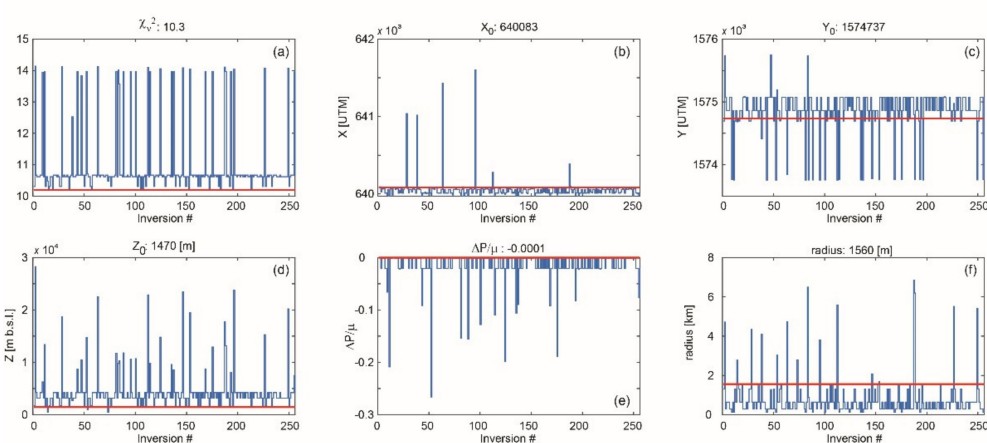

**Figure 3.** Stairs plot of the grid searches for the joint inversion of the InSAR data from orbits T300 and T321 (dataset DALLOL 2, sill source; Table 1). The inversion code implements a weighted least-squares algorithm combined with a random search grid to infer the minimum of the penalty function [31]. The red line points out the best fit solution. (**a**) penalty function $\chi_v^2$; (**b**) source location, $X_0$; (**c**) source location, $Y_0$; (**d**) source depth, $Z_0$; (**e**) dimensionless pressure change, $\Delta P/\mu$; (**f**) source radius.

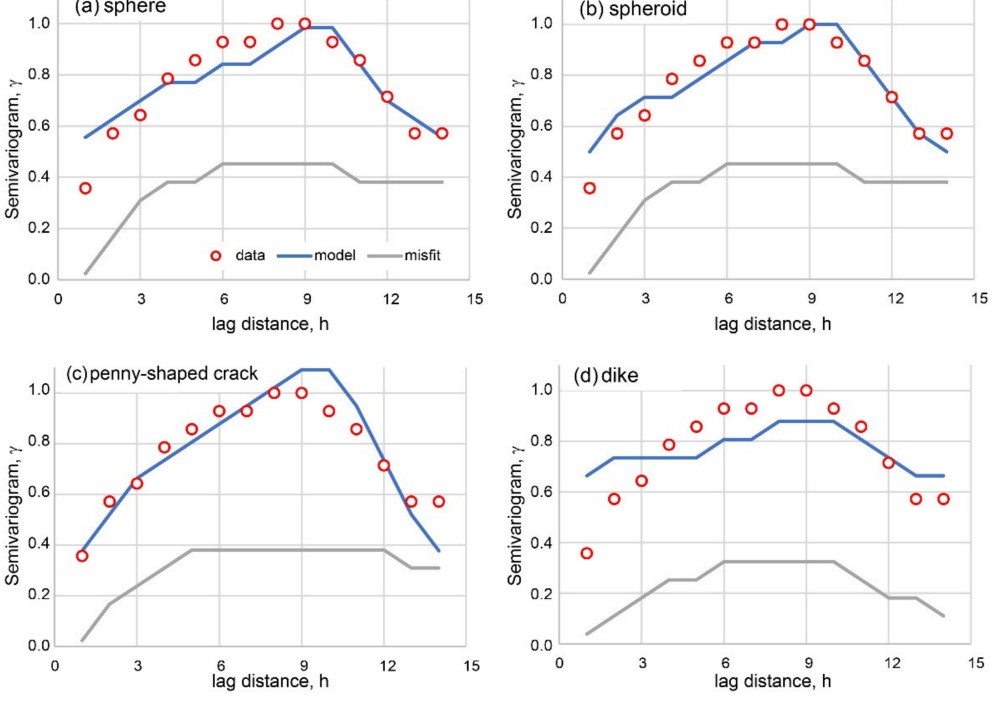

| SOURCE | Sphere | Spheroid | Penny-Shaped Crack | Dike |
|---|---|---|---|---|
| nRMSE | 0.12 | 0.10 | 0.13 | 0.20 |
| slope of misfit | 0.018 | 0.018 | 0.016 | 0.004 |

**Figure 4.** Example of semi-variograms of the deformation data (Figure 2) and models (Table 1) for the inversion of InSAR data from orbits T321 and T300 (dataset DALLOL 2). If the misfit (difference between data and model) is completely random (white noise), its variogram is a flat line (slope ~0). Full results are in Table 2.

**Table 2.** Average source parameters. Parameters are estimated from the solutions for DALLOL2 and DALLOL3 (Table 1) using a weighted average; uncertainties are the standard deviation of the weighted average. Uncertainties σ are one standard deviation.

| Source | X2v (1) | RMSE (1) | ΔX0 (2) | ±σ | ΔY0 (2) | ±σ | Depth | ±σ | Radius | ±σ | ΔV | ±σ | A | ±σ |
|---|---|---|---|---|---|---|---|---|---|---|---|---|---|---|
| | | Variogram | m | | m | | m b.s.l. | | m | | $10^6$ m$^3$/Year | | | |
| Sphere | 9.0 | 0.11 | −110 | 24 | −106 | 17 | 1232 | 63 | 380 | 230 | −0.55 | 0.07 | | |
| Spheroid | 9.0 | 0.10 | −110 | 8 | −128 | 5 | 1255 | 69 | 280 | 39 | −0.61 | 0.06 | 0.34 | 0.15 |
| Penny-shaped crack | 8.9 | 0.13 | −175 | 10 | −132 | 33 | 1357 | 199 | 1582 | 34 | −0.58 | 0.08 | | |
| Dike (3) | 10.0 | 0.17 | 215 | 281 | −164 | 210 | 589 | 71 | - | - | −0.44 | 0.08 | | |

(1) Weight given by number of pixels of the InSAR image. (2) Source offset relative to center of Dallol mountain: [640,251, 1,574,845] (UTM coordinates, 37P). (3) Location of midpoint of the dike.

The fit of the semivariograms was compared using the normalized Root Mean Square Error, *nRMSE*.

$$nRMSE = \frac{1}{\max(\gamma_d) - \min(\gamma_d)} \sqrt{\frac{\sum_{i=1}^{N}\left(\gamma_d^i - \gamma_m^i\right)^2}{N}}, \tag{1}$$

where $\gamma_d$ is the semi-variogram of the dataset, $\gamma_m$ the semi-variogram of the model and $N$ the number of pixels in the InSAR image (Table 1, Figure 4). The statistical F-test [34], usually employed to determine if the better fit of a model with more parameters is an actual improvement, is not useful in this case because of the large number of data points.

We jointly invert the InSAR velocity maps from tracks T321 (descending orbit) and T300 (ascending orbit), and tracks T321 (descending orbit) and T028 (ascending orbit) to infer the parameters of a magma body beneath Dallol volcano (Figure 2d,e; Tables 1 and 2), assuming different source geometries: a sphere, a spheroid, a horizontal penny-shaped crack, and a tensile dislocation. For the inversion, we selected the InSAR data over a 20 × 20 km area centered at Dallol volcano. However, the Dallol subsidence decays away from its center and interacts with the salt plain signal, making it difficult to identify the area where the subsidence has completely decayed to zero. To overcome this problem, we tested two different ways to minimize the noise from the salt plain: (a) we selected a reference point outside the Dallol displacement field, setting its value to be equal to the mean of the displacement outside the Dallol subsidence; (b) we masked the interferogram over the salt plain area. We obtained three data sets–the original data set (DALLOL 1), the dataset relative to the reference point outside the displacement field (DALLOL 2), and the dataset with the mask (DALLOL 3), which we inverted to infer the best fit deformation source (Table 1).

The approach described above creates six different data sets that we can use to first calibrate and then verify our models (Table 1). We calibrate our models by inverting the dataset from orbits T321–T300; we validate our models by inverting the dataset from orbits T321–028. The difference between the calibrated and verified model offers an estimate of the uncertainties in the source parameters (Table 2).

The modelling results from inverting the original dataset (DALLOL 1, Table 1) do not return a good match to the data. The solutions mostly indicate very deep sources > 10 km, which is unlikely for the shallow volcanoes of the Erta Ale ridge. These results are probably caused by the inversion fitting the signal around the plain rather than the subsidence centered at Dallol volcano. On the other hand, changing the reference point, or masking the salt plain signal, makes it possible for the inversion algorithm to fit he Dallol volcano subsidence assuming different source types (datasets DALLOL 2 and DALLOL 3). In Table 1 we show the best-fit inversion solutions for the three datasets (DALLOL 1–3) and in Table 2 the statistically significant solutions, estimated by the weighted average

of the model parameters for the DALLOL 2–3 datasets. These results clearly show that, irrespective of the geometry, the source causing the subsidence is located under the Dallol volcano at shallow depth, between 0.5 and 1.5 km, with a volume decrease between $-0.63$ and $-0.26 \cdot 10^6 \, \text{m}^3$/year (Table 1). The fit between model and data for a spherical source is shown in Figures 5–7, for illustrative purposes. Overall, the RMSE of the sphere, spheroid and penny-shaped crack are similar (Table 2), likely because of the limited number of pixels covering the central part of Dallol, which makes it difficult to discriminate between the three different geometries.

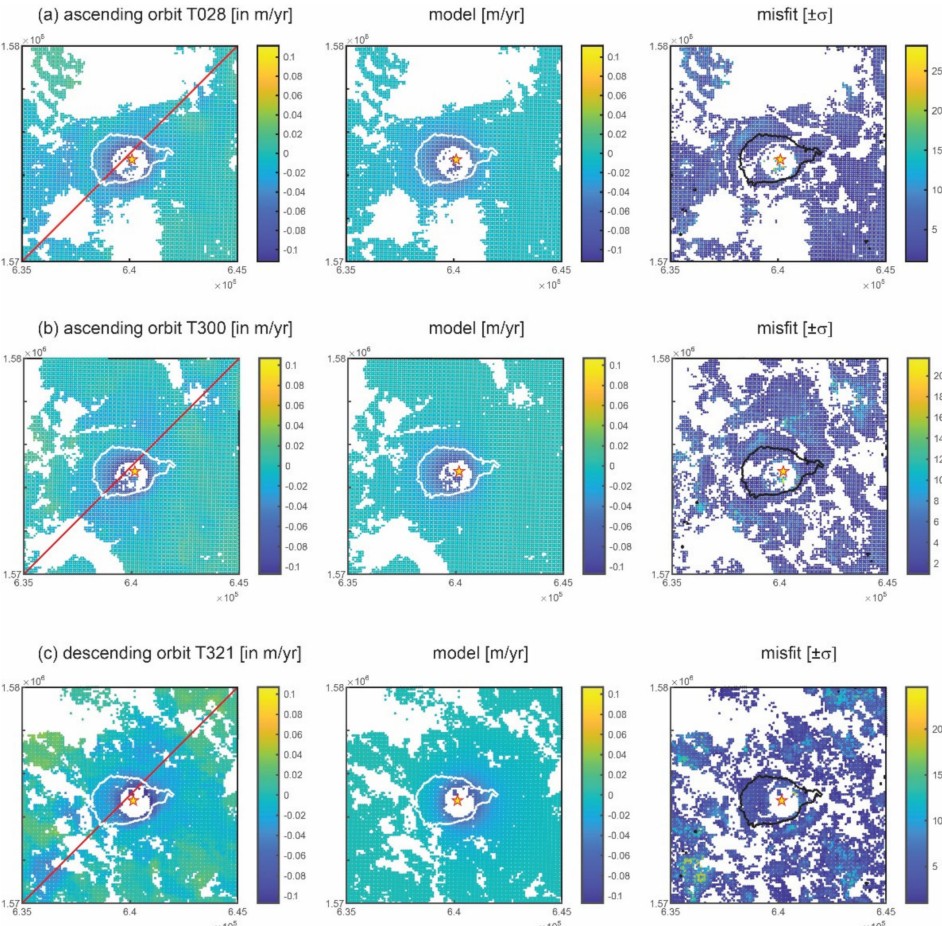

**Figure 5.** Spherical source solution: comparison between InSAR line-of-sight deformation, model (see Table 1, DALLOL 2), and misfit for the three orbits. The image coordinates are in UTM [m]. The scale of the data and model is in m/year. The scale of the misfit is based on measurement errors (5 means the misfit is five times the measurement error); all the pixels where the misfit is smaller than 2 error bars are in white. Dallol volcano is at the center of the images (white/black contour line). The red star with yellow fill is the location of the source. The red line in the data plot identifies the deformation profiles shown in Figure 6.

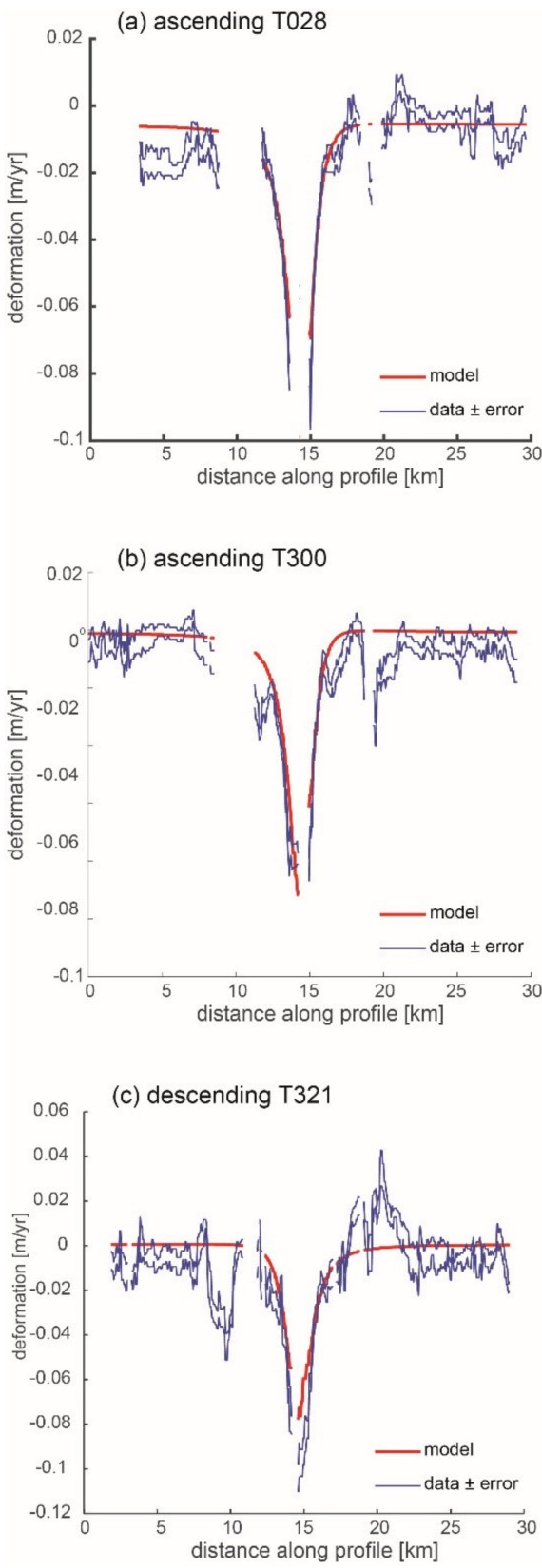

**Figure 6.** Spherical source solution for the three orbits: comparison between InSAR line-of-sight deformation and model (see Table 1; DALLOL 2) along the profile (red line) shown in Figure 5.

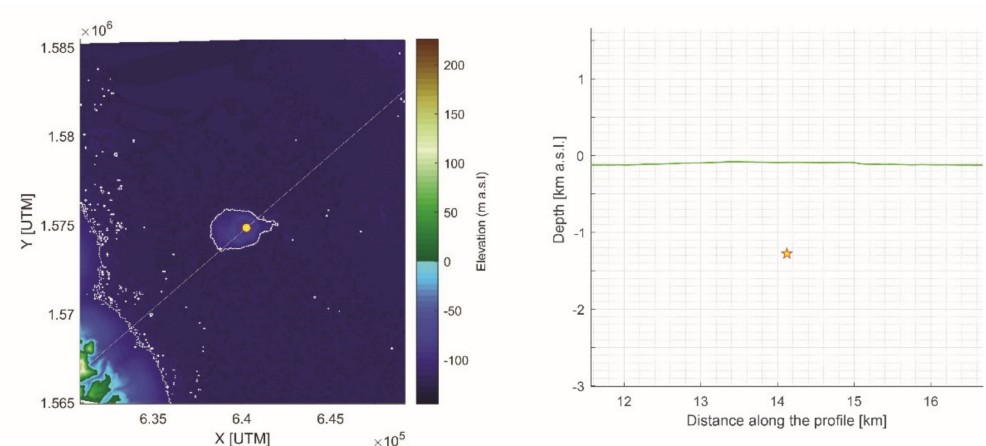

**Figure 7.** Spherical solution (Table 1, DALLOL 2). (**Left**) Source location (red circle, yellow fill); Dallol mountain is at the center of the image (white contour line); DEM from 1 Arc-Second Global SRTM (https://earthexplorer.usgs.gov/ accessed on 09 April 2021). (**Right**) Source location and depth (red star, yellow fill); the green line is the topography along the profile identified by the white diagonal line on the left panel.

### 3.2. Thermomechanical Models

In the absence of eruptions, the observed subsidence could be explained by a decrease in pore fluid pressure in a confined hydrothermal aquifer, or by thermoelastic contraction caused by the cooling of a volume of rock. The biggest challenge in investigating the cause of the subsidence is the scarcity of information on the subsurface and hydrothermal system of Dallol. Hydrothermal activity is fueled by water heated and enriched in gases by a heat source that lies primarily beneath the volcano. Stratigraphic data from the area are lacking, and the definition of the main stratigraphic units is unclear [35,36]. Because of the lack of information about the physical parameters (coefficient of linear thermal expansion, poroelastic expansion coefficient, thickness of the hydrothermal aquifer) of the Dallol geothermal system, we employed the parameters from a natural analog, the Hellisheidi geothermal area in Iceland. Hellisheidi has similarities with Dallol since it is a basaltic volcano with a hydrothermal field in a divergent tectonic setting ([37]; Table 3). The Hellisheidi's geothermal field experienced subsidence from a source that is modelled by a penny-shaped crack at a depth of 1.3 km and with a radius of 1.6 km [37].

In simple models for poroelastic and thermoelastic deformation, the source strength for a deforming cavity is given by [38].

Poroelastic deformation:

$$s = (1 - \nu)\frac{c_f V_f \Delta P_f}{\pi} \tag{2}$$

Thermoelastic deformation:

$$s = (1 - \nu)\frac{\alpha_t V_t \Delta T}{\pi} \tag{3}$$

Deformation from the geodetic volume change:

$$s = (1 - \nu)\frac{\Delta V}{\pi} \tag{4}$$

where $\nu$ is the Poisson's ratio, $\Delta V$ is the geodetic volume change, $c_f$ is Geertsma's uniaxial poro-elastic expansion coefficient, $V_f$ is the volume of the hydrothermal aquifer experiencing the pressure change $\Delta P_f$, $\alpha_t$ is a coefficient of linear thermal expansion, equivalent to one third of the volumetric thermal expansion coefficient, and $V_t$ is the volume of rock experiencing the temperature change $\Delta T$. Equations (2)–(4) are exact for a spherical source but only an approximation for other source geometries. Since we have estimated the source

geodetic volume change $\Delta V$, the radius $a$ and their uncertainties $\sigma_{\Delta V}$ and $\sigma_a$ (Table 2), we can use Equations (2)–(4), and error propagation to derive estimates of the changes in the poro-elastic pressure change $\Delta P_f$ and thermoelastic temperature change $\Delta T$ and their uncertainties (Table 3)

$$\Delta P_f = \frac{\Delta V}{c_f V_f} \quad \sigma_{\Delta P}^2 = \left(\frac{1}{c_f V_f}\right)^2 \sigma_{\Delta V}^2 + \left(\frac{\Delta V}{c_f^2 V_f}\right)^2 \sigma_c^2 + \left(\frac{\Delta V}{c_f V_f^2}\right)^2 \sigma_V^2 \ , \tag{5}$$

$$\Delta T = \frac{\Delta V}{\alpha_t V_t} \quad \sigma_{\Delta T}^2 = \left(\frac{1}{\alpha_t V_t}\right)^2 \sigma_{\Delta V}^2 + \left(\frac{\Delta V}{\alpha_t^2 V_t}\right)^2 \sigma_\alpha^2 + \left(\frac{\Delta V}{\alpha_f V_f^2}\right)^2 \sigma_V^2 \ , \tag{6}$$

where $\sigma$ indicates the uncertainties of the parameters. The volume $V_f$ and $V_t$ are approximately

$$V_f \approx V_t \approx \pi a^2 h, \quad \sigma_V^2 \approx (2\pi a h \sigma_a)^2 + \left(\pi a^2 \sigma_h\right)^2 \quad \text{sill} - \text{like} \tag{7}$$

where $h$ is the thickness of the penny-shaped crack. The estimates for pressure and temperature decreases are in Table 3.

**Table 3.** Estimate of poro-elastic pressure and thermoelastic temperature changes for a penny-shaped crack, sub-surface structure. Errors are one standard deviation. Parameters for the hydrothermal system are from Hellisheidi volcano, a possible natural analog [37].

| Parameter | Source | Value | Error | Units | | | |
|---|---|---|---|---|---|---|---|
| $a$ | Table 2 | 1582 | 34 | m | | | |
| $h$ | [39] | 750 | 250 | m | | | |
| $V_f, V_t$ | | 5.9 | 2.0 | km$^3$ | | | |
| $\Delta V$ | Table 2 | $-0.58$ | 0.08 | $10^6$ m$^3$ | | | |
| | *Poroelastic medium* | | | | *Thermoelastic medium* | | |
| **Parameter** | **Value** | **Error** | **Units** | **Parameter** | **Value** | **Error** | **Units** |
| $c_f$ | 4.5 | 3.0 | $10^{-10}$ Pa$^{-1}$ | $\alpha_t$ | 1.0 | 0.5 | $10^{-5}$ K$^{\circ -1}$ |
| $\Delta P^f$ | $-0.22$ | 0.17 | MPa | $\Delta T$ | $-10$ | 6 | K$^\circ$ |

$\alpha_t$ after [39]; $c_f$, $h$ after [37].

## 4. Discussion

InSAR shows that linear subsidence started at Dallol volcano in October 2008 and continued at least until early 2010, when the ENVISAT acquisitions in the area stopped. More recent Sentinel-1 observations show that the Dallol subsidence was still ongoing at least until March 2015 [40,41]. The Dallol subsidence between October 2008 and February 2010 is best fit by a radially symmetric magma chamber at 0.5–1.5 km depth at 66% confidence interval, experiencing a volume decrease between $-0.63$ and $-0.26$ $10^6$ m$^3$/year. Our best-fit depth for the Dallol magma chamber is consistent with a previous InSAR study of the 2004 dyke-induced subsidence which placed the Dallol chamber between 1.5–3.3 km of depth [15]. Although the difference in magma chamber depths between the two InSAR studies is not significant, our model places the deformation source at the roof of the source inferred by [15].

Minor cooling and contraction, or minor depressurization, of the roof in the Dallol chamber is a reasonable explanation for the observed ground subsidence, but the reason why the subsidence suddenly started in October 2008 is less obvious.

An important factor to consider is the role of volatiles. Magma rich in exsolved volatiles is highly compressible to the point that the compression of residing magma may have offset the pressurization caused by magma inflow, resulting in no ground inflation prior to 2008 [42–44]. Outgassing of volatiles from the roof of the chamber [45] agrees with the somewhat shallower source depth inferred for the 2008–2010 Dallol subsidence

compared to the 2004 dike injection that tapped the Dallol magma system. The intense Dallol hydrothermal field could be responsible for the observed subsidence since we estimate that only minor drops in pressure and temperature could explain the observed deformation (Table 3). Alternative explanations like (a) magma draining from a shallow crustal chamber into the rift axis, (b) crustal thinning coupled with loading by the volcano and dyke intrusions and weakening of the crust by heating [46], or (c) a deep connection with other active volcanoes in the Erta Ale region, are less likely. The extensional tectonic regime in northern Afar did not change in 2008 when subsidence suddenly started, nor is Dallol volcano likely to exert much loading, as it is only a small 50 m high mountain. The onset of the Dallol subsidence in October 2008 nearly coincides with the Alu-Dalafilla eruption along the Erta Ale magma in November 2008, but it is unlikely that a deep magma or pressure connection exists between volcanoes over 50 km apart and in any case no deformation in the area between the volcanoes was observed. Earthquakes recorded in the Erta Ale region by a local network 2005–2009 showed a clear seismicity increase in November 2008 confined to the eruptive site of Alu-Dalafilla.

## 5. Conclusions

InSAR observations show that a magmatic/hydrothermal system is active beneath the Dallol hydrothermal area. Inverse modeling of InSAR velocity maps demonstrate that the system is shallow, like the other active volcanoes of the Erta Ale ridge. Our new observations together with a previous InSAR study of a dike injection and seismicity along the Dallol rift indicate that Dallol is an active magmatic rift segment with a magma chamber and a hydrothermal field. The most likely explanations for the subsidence of Dallol volcano is a combination of outgassing (depressurization), and cooling and contraction of the roof of a shallow crustal magma chamber and its hydrothermal system (Table 3). Although it may seem unlikely that this would start so quickly in 2008, there are examples of deformation abruptly switching to subsidence in other hydrothermal areas, such as the Norris Geyser Basin area in Yellowstone caldera [47].

**Author Contributions:** Conceptualization, M.B. and C.P.; methodology, M.B. and C.P.; software, M.B.; validation, M.B.; formal analysis, M.B. and C.P.; investigation, S.M.; data curation, C.P.; writing—original draft preparation, M.B. and C.P.; writing—review and editing, M.B. and C.P.; supervision, M.B. and C.P.; funding acquisition, M.B. All authors have read and agreed to the published version of the manuscript.

**Funding:** This research was funded by Sapienza–University of Rome, Piccoli Progetti Universitari 2015 and 2020, USAID via the Volcano Disaster Assistance Program and the U.S. Geological Survey Volcano Hazards Program.

**Data Availability Statement:** Deformation data from InSAR and modeling software are available from the authors. ENVISAT data are freely and openly available from the ESA website https://earth.esa.int/eogateway/missions/envisat/data, (accessed on 9 April 2021).

**Acknowledgments:** Comments by Mike Poland (USGS), Mike Clynne (USGS) and two anonymous reviewers greatly helped to improve the manuscript. The original version of dMODELS is available online at https://pubs.usgs.gov/tm/13/b1/, (accessed on 9 April 2021). Any use of trade, firm, or product names is for descriptive purposes only and does not imply endorsement by the U.S. Government.

**Conflicts of Interest:** The authors declare no conflict of interest.

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
