# Peer review of "The 2008–2010 Subsidence of Dallol Volcano on the Spreading Erta Ale Ridge: InSAR Observations and Source Models"

_remotesensing, doi:10.3390/rs13101991_

Round 1

Reviewer 1 Report

The article presented comprehensively analyzes the subsidence induced by the Dallol volcano in Ethiopia. The description provided is structured in a scrupulous way highlighting in a precise manner the results obtained by means of the InSAR deformation measures. In this circumstance, the paper can be accepted for publication, however clarification is required:

  • Q1: Given the article, it would be advisable to avoid indicating in the various sections the redundant wording "We analyze", "We invert" etc ... since it is understood that the same authors have provided, introduced and discussed the proposed theme. Please review;
  • Q2: Please make the set of Figures 3 and 4 visible, it looks too blurry and makes reading difficult;
  • Q3: Please better formulate the period between 226 to 231 is hostile to understand. Also, after the reference [39], line 228, it should be put "." and not ",".
  • Q4: The innovative contribution of the proposed methodology is not clear. Provide more details on the peculiarities of the method.

Author Response

First, we would like to thank the reviewer for his/her feedback and comments. We appreciated that he/she took time from his/her busy schedule to review our manuscript. 

  • Q1: Given the article, it would be advisable to avoid indicating in the various sections the redundant wording "We analyze", "We invert" etc ... since it is understood that the same authors have provided, introduced and discussed the proposed theme. Please review;

DONE

  • Q2: Please make the set of Figures 3 and 4 visible, it looks too blurry and makes reading difficult;

Done. We improved the quality of the JPEG files of Figure 3 and Figure 4 to 600DPI. We update all the Figures in the paper to 600 DPI

  • Q3: Please better formulate the period between 226 to 231 is hostile to understand. Also, after the reference [39], line 228, it should be put "." and not ",".

Done

  • Q4: The innovative contribution of the proposed methodology is not clear. Provide more details on the peculiarities of the method.

Done. Lines 183-195

In particular (lines 191 - 195)

The approach described above creates six different data sets that we can use to first calibrate and then verify our models (Table 1). We calibrate our models by inverting the dataset from orbits T321-T300; we validate our models by inverting the dataset from orbits T321-T028. The difference between the calibrated and verified model offers an estimate of the uncertainties in the source parameters (Table 2).

Because of the limited data sets, the standard approach in volcano geodesy is to calibrate the model. Models are not verified. We took advantage of the available dataset to both calibrate and the verify the model.

Reviewer 2 Report

This paper is not a good fit into Remote Sensing, as there is no relevant innovation in terms of Remote Sensing. I would suggest the authors to find a journal that would better fit, so that they can find the readers this paper deserves.

Nevertheless, the paper is interesting and I believe of interest to readers from e.g. geology. 

The structure of the manuscript should change to fit into the structure of remote sensing. The results section is mixed with the methodology and this should be separated.

The introduction should also put the methodologies used into the context of remote sensing's state of the art. Currently the introduction introduces the geological background, but not the remote sensing background. This is also  relates to my initial comment on that this is not a remote sensing paper to begin with. To make it a better fit, the authors could at least introduce the scientific background of the remote sensing methodology used in the paper.

Author Response

First, we would like to thank the reviewer for his/her feedback and comments. We appreciated that he/she took time from his/her busy schedule to review our manuscript. 

This paper is not a good fit into Remote Sensing, as there is no relevant innovation in terms of Remote Sensing. I would suggest the authors to find a journal that would better fit, so that they can find the readers this paper deserves.

We disagree with the reviewer comment. This is a paper on the application of remote sensing to monitoring of volcanic unrest. The journal has published many papers with a similar flavor (e.g., the special number where we submitted this paper)

Nevertheless, the paper is interesting and I believe of interest to readers from e.g. geology. 

The structure of the manuscript should change to fit into the structure of remote sensing. The results section is mixed with the methodology and this should be separated.

We disagree with the reviewer. In our opinion, a rigid division of the paper in the standard Introduction, Methods (InSAR, Modelling), Results (InSAR, Modeling), Discussion would have made difficult to read and appreciate our work. We prefer a structure where emphasis is given to the topics 

The introduction should also put the methodologies used into the context of remote sensing's state of the art. Currently the introduction introduces the geological background, but not the remote sensing background. This is also  relates to my initial comment on that this is not a remote sensing paper to begin with. To make it a better fit, the authors could at least introduce the scientific background of the remote sensing methodology used in the paper.

Done. See lines 61 to 71 of the revised version